# SPInDel Analysis of the Non-Coding Regions of cpDNA as a More Useful Tool for the Identification of Rye (Poaceae: *Secale*) Species

**DOI:** 10.3390/ijms21249421

**Published:** 2020-12-10

**Authors:** Lidia Skuza, Ewa Filip, Izabela Szućko, Jan Bocianowski

**Affiliations:** 1Institute of Biology, University of Szczecin, 13 Wąska, 71-415 Szczecin, Poland; ewa.filip@usz.edu.pl (E.F.); dyrektor.bio@usz.edu.pl (I.S.); 2The Centre for Molecular Biology and Biotechnology, University of Szczecin, 13 Wąska, 71-415 Szczecin, Poland; 3Department of Mathematical and Statistical Methods, Faculty of Agronomy and Bioengineering, Poznań University of Life Sciences, 28 Wojska Polskiego, 60-637 Poznań, Poland; jan.bocianowski@up.poznan.pl

**Keywords:** *Secale*, cpDNA, mtDNA, SPInDel, hypervariable regions

## Abstract

*Secale* is a small but very diverse genus from the tribe *Triticeae* (family *Poaceae*), which includes annual, perennial, self-pollinating and open-pollinating, cultivated, weedy and wild species of various phenotypes. Despite its high economic importance, classification of this genus, comprising 3–8 species, is inconsistent. This has resulted in significantly reduced progress in the breeding of rye which could be enriched with functional traits derived from wild rye species. Our previous research has suggested the utility of non-coding sequences of chloroplast and mitochondrial DNA in studies on closely related species of the genus *Secale*. Here we applied the SPInDel (Species Identification by Insertions/Deletions) approach, which targets hypervariable genomic regions containing multiple insertions/deletions (indels) and exhibiting extensive length variability. We analysed a total of 140 and 210 non-coding sequences from cpDNA and mtDNA, respectively. The resulting data highlight regions which may represent useful molecular markers with respect to closely related species of the genus *Secale*, however, we found the chloroplast genome to be more informative. These molecular markers include non-coding regions of chloroplast DNA: *atpB-rbcL* and *trnT*-*trnL* and non-coding regions of mitochondrial DNA: *nad1B*-*nad1C* and *rrn5/rrn18*. Our results demonstrate the utility of the SPInDel concept for the characterisation of *Secale* species.

## 1. Introduction

Rye (*Secale cereale* L.) is commonly grown in the fields of Northern and Eastern Europe. *S. cereale* L. is highly tolerant to diverse environmental stresses, as drought and frost [1,2]. It contains genes conferring biotic stress resistance against various diseases (e.g., stem rust, leaf rust, and yellow rust [3] or powdery mildew [4]. In view of that the wild and weedy forms may crossbreed with cultivated rye [5], these taxa, along with the landraces, constitute gene pools for desirable genes. They can be regarded as genetic resource reservoirs for new niches and future breeding programs of wheat, triticale and other crops [6].

In spite of the high economic value of rye, its huge genome size (2*n* = 2 × = 14, 1Cx ~ 7.9 Gbp) [7] (the largest among the diploid species of Poaceae family) and highly repetitive genome 92% [8] have prevented the utilization of its genome information for crop improvement. Hence understanding genetic structuring of the genus *Secale* and distribution of genetic diversity within the genus is extremely important.

Unfortunately, the phylogenetic relationships within the genus *Secale* remain unclear. A division of the genus *Secale* into 15 different species has even been adopted [9,10], while Frederiksen and Petersen [11,12] recognized only three *Secale* species: *Secale sylvestre, Secale strictum* and *S. cereale*. According to the newest classification [13] the genus *Secale* is classified into four species: *S*. *cereale*—an annual allogamous species, *S*. *sylvestre* and *vavilovii*—an annual autogamous species and *S*. *strictum*—a perennial wild-type allogamous species [14]. Moreover, *S. cereale* also comprises eight subspecies, *S. strictum*—5. *S. cereale* ssp. *cereale* is the only cultivated species, although *S. strictum* may have been used as a forage crop [15]. Probably the most distant species is *Secale sylvestre* [15,16,17,18,19,20,21,22,23] due to its low crossability with other species, the lowest amount of t-heterochromatin [16] and the smallest genome (7.23 pg) [17]. For rye breeding *S. vavilovii* is attractive. It is characterized by high self-fertility, sprouting and sterilising cytoplasm, high protein content, resistance to diseases such as fusarium ear blight, septoria leaf blotch. Chromosomes of *S. strictum* are sources for resistance to yellow rust, Russian wheat aphid, grain hardness, increased protein and arabinoxylan content.

A number of studies on rye genetic diversity have been conducted using different marker systems, which include AFLP (Amplified Fragment Length Polymorphism) [24], SSR (Simple Sequence Repeats) [16,25,26,27,28,29], DArT (Diversity Arrays Technology) [30,31], and recently SNPs (Single nucleotide polymorphism) [32] and GWAS (Genome-wide association study) [33]. These markers tested individually or as a set of loci were unsuccessful in the study of the genetic variability of the genus *Secale*. The conducted studies only confirmed the lack of monophyletism in the *Secale* sp. subspecies, which resulted from a similar inter- and intraspecific distance. This proves that gene flow continues between rye species and the recent evolution, which does not allow the full formation of isolation mechanisms [34].

The results of several studies on the genus *Secale* indicate non-coding sequences as a source of markers for the differentiation of rye species. The non-coding sequences evolves faster than the coding region and due to this provides more information for phylogenetic analysis because of the higher variation rate [35,36,37]. The SpinDel program is often used for this type of analysis.

The theoretical basis of SPInDel (Species Identification by Insertions/Deletions) is the analysis of hypervariable regions with variable lengths having a large number of insertions/deletions (indels) interspersed with significantly conserved regions (Figure 1). A unique numeric profile for each species is defined by the combination of fragment lengths, allowing its identification.

Similarly, our previous research, which relied on analyzes of the coding and non-coding sequences of the mitochondrial, chloroplast and nuclear genomes, supports these results [38]. These studies demonstrated a better suitability of cpDNA for the analysis of *Secale* species. This has also been confirmed by several other authors in other land plants [39,40,41,42,43,44,45]. The cpDNA of plants is particularly suitable for the application of the SPInDel concept by having several coding regions (usually conserved) interspersed with large non-coding domains such as introns or intergenic spacers (usually rich in indels). Unfortunately, there are no data on the mitochondrial genome so far. Here we tested the use of the SPInDel concept for the identification of 35 accessions of the genus *Secale*, representing 13 most often distinguished species and subspecies, originating from various seed collections in the world. We also checked which organelle genome is more informative for the genus *Secale*.

## 2. Results

### 2.1. Analysis of Variable-Length cpDNA Sequences

A total of 3031 bp from 140 sequences were analyzed. The species and characteristic of each target region is described in Appendix A.

The potential use of SPInDel profiles for species identification purposes requires the existence of ‘species-specific SPInDel profiles’: those that are only found in one species within a taxonomic group and allow their unequivocal identification. The mean number of species-specific profiles (Nsp) was 2.5 (from 1 to 4), while the mean number of species with shared profiles (N(species)sh) was 3.25 (varied from 2 to 4) (Table 1), i.e., almost all regions in this group had specific profile. If all profiles were specific, Nsp will be equal to the number of different profiles (N). If some profiles are shared, then N > Nsp. A profile can be shared between two or more species, therefore the number of species with shared profile can be higher than the number of species-shared profiles.

The frequency of species-specific profiles is 1.00, indicating that all species have a unique SPInDel profile. The mean frequency of species-specific profiles (f_sp_) was 0.83, ranging from 0.72 to 0.91 (Table 1). This result suggests that all cpDNA regions had almost unique combination of fragments lengths.

All species from the *Secale* genus have different profiles, and the average numbers of pairwise differences among hypervariable ranging from 0.44 (*S. s.* ssp. *africanum*; *atpB-rbcL* and *S. c.* ssp. *afghanicum*; *trnT*(UGU)-*trnL*(UAA)) to 4.17 (*S. c.* ssp. *segetale*; *trnL*(UAA) (Appendix A). The discrimination of all species is possible with maximum four hypervariable regions (*S. sylvestre atpB-rbcL*; *S. c.* ssp. *ancestrale*, *S. c.* ssp. *segetale*, *S. s.* ssp. *kuprijanovii trnL*(UAA); *S. s.* ssp. *anatolicum trnD-trnT*) (Appendix A). Most species have one hypervariable region: six species in 3 cpDNA regions and five in 1 (*trnD-trnT*) (Appendix A), average from 1.77 in *atpB-rbcL* to 2.08 in *trnD*[tRNA-Asp(GUC)]-*trnT*[tRNA-Thr(GGU)] (Appendix A).

If we consider species analysis, the mean number of species-specific profiles (Nsp) was 11.69 (from 4 to 16), and the mean number of species with shared profiles (N(species)sh) was 0 (Appendix A), i.e., no species in this group had shared profile. All analyzed sequences were species specific, whether or not they represented a subspecies group, therefore the mean frequency of species-specific profiles (f_sp_) was 1.

The four non-coding cpDNA regions from 35 accessions were concatenated. The maximum frequency of species-specific profiles of the concentration (f_sp_ = 0.35) was reached with the use of hypervariable regions *trnT*(UGU)-*trnL*5’exon (Figure 2A). The average number of pairwise differences for the concatenated regions was 7.73, a value close to the maximum that can be obtained with four hypervariable regions. A total of 455 pairwise comparisons yielded differences in the four hypervariable regions, while 384 cases (84%) had four differences. Only 116 cases were different by less than two hypervariable regions (Figure 2B). Figure 2C shows the ‘region by region’ analysis for the concatenated for cpDNA. The regions *trnL*(UAA) intron vs. *trnD*[tRNA-Asp(GUC)]-*trnT* [tRNA-Thr(GGU)] had the highest average pairwise differences, with *p* = 0.73.

### 2.2. Analysis of Variable-Length mtDNA Sequences

A total of 4.171 bp from 210 sequences were analyzed. The species and characteristic of each target region is described in Appendix A.

The mean number of species-specific profiles (Nsp) was 2.67 (from 1 to 7), while the mean number of species with shared profiles (N(species)sh) was 2.17 (varied from 1 to 4) (Table 1). The mean frequency of species-specific profiles (f_sp_) was 0.73, ranging from 0.45 to 0.89 (Table 1). i.e., almost all regions in this group had specific profile.

If we consider species analysis, the mean number of species-specific profiles (Nsp) was 17.54 (from 6 to 24), and the mean number of species with shared profiles (N(species)sh) was 0 (Appendix A), i.e., no species in this group had shared profile. All analyzed sequences were species specific, whether or not they represented a subspecies group, therefore the mean frequency of species-specific profiles (f_sp_) was 1.

All species from the *Secale* genus have different profiles, and the average numbers of pairwise differences among hypervariable ranging from 0.42 (*S. s.* ssp. *ciliatoglume*; *nad4/1-2*) to 5.54 (*S. c.* ssp. *ancestrale*; *rps12-2/nad3-1*) (Appendix A).

The discrimination of all species is possible with maximum seven hypervariable regions (*S. s.* ssp. *kuprijanovii*, *S. vavilovii*; *nad1*exonB-*nad1*exonC) (Appendix A).

Most species have more than one hypervariable region (about 74%), average from 2.62 in *nad4/1-2* to 3.69 in *nad1*exonB-*nad1*exonC (Appendix A).

The six non-coding mtDNA regions from 35 accessions were concatenated. The maximum frequency of species-specific profiles of the concentration for mtDNA (f_sp_ = 0.23) was reached with the use of hypervariable regions *rps12-1/nad3-2*, *rps12-2/nad3-1* and *rrn5/rrn18-1* (Figure 3A). The average number of pairwise differences for the concatenated regions was 6.61, a value close to the maximum that can be obtained with five hypervariable regions. A total of 595 pairwise comparisons yielded differences in the five hypervariable regions, while 455 cases (76%) had three differences. Only 122 cases were different by less than two hypervariable regions (Figure 3B). Figure 3C shows the ‘region by region’ analysis for the concatenated for mtDNA. The regions *nad*4*L-orf*25 vs. *rps12-1/nad3-2* and *nad4/1-2* vs. *nad4L-orf25* had the highest average pairwise differences, with p equal, respectively, 0.72 and 0.71.

## 3. Discussion

The cpDNA and mtDNA regions are widely used as markers in phylogenetic and phylogeographic studies [46,47]. Additionally, for most species, including rye, the data is incomplete. So far, *S*. *cereale* chloroplast genome sequence has been published [48], while rye mtDNA has not been fully sequenced.

The cpDNA appears to have much more stable structure than the mitochondrial genome of plants in case of intramolecular rearrangement. The plastid genome substitution rate is 3–4 times higher than that of plant mtDNA [49]. In cpDNA noncoding regions most of the variability observed concerns insertion-deletion (indel) mutations. As stated by other authors, however, they should be treated with caution as they may indicate heteroplasmy [50]. Nevertheless, indels were analyzed in the following studies due to the fact that they were found to be common and often phylogenetically informative [51,52,53].

The cpDNA shows a much more stable structure in case of intramolecular rearrangement than the mitochondrial genome of plants. The rate of plastid genome substitution is 3–4 times higher than that of plant mtDNA [49]. Most of the variability observed in cpDNA noncoding regions concerns insertion-deletion (indel) mutations, but, as stated by other authors, they should be treated with caution as they may indicate heteroplasmy [50]. However, indels were analyzed in the following studies due to the fact that they were found to be common and often phylogenetically informative [51,52,53].

The multiple insertions and deletions (indels), which are present in the target genomic regions is generally regarded as a problem, as it introduces ambiguities in sequence alignments. However, in a few recent works a high level of species discrimination is attainable in all taxa of life by considering the length of hypervariable regions defined by indel variants has been shown [54]. Each species is tagged with a numeric profile of fragment lengths.

Indels are less prone to back and recurrent mutations, and thanks to that the probability of misclassifications is greatly reduced. Genomic regions with have multiple indels was used for species-identification procedures with high efficiency not only in plants chloroplast *trnL*(UAA) intron [55], but also in bacteria [56], fungi [57] and animals [58,59]. Unfortunately, there is no such information on the species of rye.

For indel detection several software tools are available [60,61]. In the present study a multifunctional computational workbench (named SPInDel for SPecies Identification by Insertions/Deletions) was used. The SPInDel approach relies on the analysis of multiple loci, which presents a clear advantage over methods targeting a single locus, therefore the occurrence of intraspecific or intraindividual variability in hypervariable regions does not pose serious problems [62,63,64]. A correct identification is still possible based on the information from the remaining loci even in cases where one (or more) SPInDel hypervariable region(s) have a different from the reference length.

In our previous studies the phylogenetic indels context were not analysed because algorithms describing substitution models are not able to model indel-type changes and they are removed by tree-creating programs. However, basic data on length and number of indels were obtained [38]. The number of identified indels in cpDNA was not very high compared to the results presented by other authors [65]. While the number of identified indels in mtDNA was comparable to the results of other authors in mtDNA of plants [66,67].

The occurrence of deletions and insertions as well as numerous nucleotide substitutions is a common phenomenon in the *atpB-rbcL* region. The majority of non-coding regions rich in these base pairs show a low number of functions [68]. SPinDel analysis of this regions showed the highest frequency of species-specific profiles (f_sp_ = 0.91) (Table 1), which means that almost all species have a unique profile.

The region *trnT*(UGU)-*trnL*(UAA)5′exon shows a high frequency of insertions or deletions, depending on the species, which makes it possible to use them as genetic markers [38].

The mean f_sp_ of the species analysed for this region (f_sp_ = 0.83) was very close to the mean f_sp_ for the *atpB-rbcL* region (Table 1).

One of analysed regions, the chloroplast *trnL*(UAA) intron, is known for its potential as species-specific marker due to low intra- and higher inter-specific genetic variation [32,55]. Our results show that *trnL* does not represent the most variable non-coding region of chloroplast DNA (Table 1).

We detected maximum diversity values in the intraspecific data sets for all target regions (Figure 2C). This is the first analysis performed using the SPInDel program, indicating the usefulness of the analyzed regions in *Secale* species.

Sequences from these cpDNA regions are often used in combination with other sequences in order to obtain additional data and a better resolution for phylogenetic studies [37,69,70,71,72]. Our results also showed that the regions *trnL*(UAA) intron vs. *trnD*[tRNA-Asp(GUC)]-*trnT*[tRNA-Thr(GGU)] had the highest average pairwise differences (Figure 2B). Moreover, the *tr*n*D*[tRNA-Asp(GUC)]—*trnT*[tRNA-Thr(GGU)] region in our previous studies showed the greatest variability of rye species compared to the entire pool of analyzed cpDNA regions, which predisposed it to study closely related species.

Indels polymorphisms have a sufficiently rapid evolutionary rate of accumulation despite the low intra-specific diversity in cpDNA genes, that allows for discrimination between closely related taxa [73]. The frequency of species-specific SPInDel cpDNA profiles reached the maximal possible value (f_sp_ = 1) in the intraspecific level (Appendix A) and 0.91 for the *atpB-rbcL* region (Table 1). The real potential of the SPInDel concept was revealed by the concatenation of the cpDNA target regions (Figure 2, Table 1). The results of this study confirmed this, because they focus on the high variability of the studied regions of the chloroplast genome in the majority of taxa (Table 1). The *trnT*(UGU)-*trnL*(UAA)5′exon and *trnD*[tRNA-Asp(GUC)]-*trnT*[tRNA-Thr(GGU)] regions were used to the greatest extent for the analysis of closely related species species, which gave the best results in combination.These regions should be a useful tool as molecular markers for the study of closely related species, especially at the interspecies level of the genus *Secale*.

The mitochondrial genomes of plants are characterized by the presence of a relatively large number of group II introns compared to fungal and baterial mtDNA [74,75]. Group I introns, which are located in the *coxI* gene, are also contained in several plant genera, including *Peperomia* and *Marchantia* [76]. However, there is no correlation between phylogenesis and the presence of this intron, indicating that it was introduced by horizontal gene transfer. Fungal species were probably the donor.

In our previous study, a total of 45 indels in mtDNA have been identified [38]. It is comparable to the results of other authors in plant mtDNA. Rye mtDNA sequence data are not available. Only the winter wheat (*Triticum aestivum* cv. Chinese Yumai) mtDNA sequence was was published and was found to be very similar to the spring wheat sequence (*T. aestivum* cv. Chinese Spring) [77]. We determined 0 to 40 (in nad*4/1-2*) indels with sizes ranging from 1 bp to 5 bp in the *rps12-nad3*(2) and *nad4/1-2* regions, respectively [38].

Mitochondrial *nad1B*-*nad1C* region, which is located in exon b and c [78], has highly conserved nature whithin this group of introns [38]. Nevertheless, the *nad1* intron region may serve as a useful molecular marker in population studies. SPinDel analysis of this regions showed the highest frequency of species-specific profiles (f_sp_ = 0.89) (Table 1), meaning that almost all species have a unique profile.

The region located within subunit 4 of the *nad4* gene is considered to be a slowly evolving mitochondrial marker. Its evolution occurs 23 times slower than that of the ITS rDNA sequence [79]. Thus, it could be a useful insightful tool in deliberating on phylogenetic relationships. The research has shown that among all tested mtDNA sequences *nad4/1-2* region proved to be the most informative [38]. Unfortunately, our analysis showed that frequency of insertions or deletions is not very high (f_sp_ = 0.72) (Table 1).

Another analyzed intergenic region, *nad4L-orf25* shows the lowest frequency of species-specific profiles (f_sp_ = 0.45) among all analyzed regions, both mtDNA and cpDNA (Table 1). This confirms our earlier research [38] and the data on sugar beet [80].

Two different combinations of primers were used to amplify the intergenic sequences of the *rps12-nad3* region: the first amplified only the intergenic region, the second–the intergenic region and the *nad3* gene. These regions were described as variable regions, while the first region proved conserved in *Secale* species and subspecies [38]. Similarly, SPinDel analysis of this regions showed great differences between them: frequency of species-specific profiles was lower in the *rps12-1/nad3-2* region than in the *rps12-2/nad3-1* (f_sp_ = 0.67and f_sp_ = 81, respectively) (Table 1).

The mean f_sp_ of the species analysed for this the *rrn5-rrn18* region (f_sp_ = 0.86) was very close to the mean f_sp_ for the *nad1B*-*nad1C* region (Table 1). It was confirmed that sequences of this regions proved to be the most informative among all tested mtDNA sequences.

The frequency of species-specific SPInDel cpDNA profiles reached the maximal possible value (f_sp_ = 1) in the intraspecific level (Appendix A) and 0.91 for the *atpB-rbcL* region (Table 1).

The concatenated analysis showed slightly lower frequency of species-specific profiles of the concentration for mtDNA (f_sp_ = 0.23), than for cpDNA (f_sp_ = 0.35) (Figure 2A). The maximum frequency was reached with the use of 3 regions *rps12-1/nad3-2*, *rps12-2/nad3-1* and *rrn5/rrn18-1* (Figure 3A). The average number of pairwise differences for the concatenated regions was also lower compared to cpDNA (6.61 and 7.73, respectively). The concatenated mtDNA analysis of other values were also lower than in cpDNA (Figure 2B,C), despite the greater number of analyzed regions.

## 4. Materials and Methods

The plant material consisted of 35 accessions of the genus *Secale*, 13 cultivated and non-cultivated species and subspecies of rye. They were obtained from several world collections (Center for Biological Diversity Conservation in Powsin—Warsaw, Poland; United States Department of Agriculture—Agricultural Research Service in Beltsville, Maryland, USA; Nordic Genetic Resource Center in Alnarp, Sweden). The list of species, as well as the accession numbers for each sample, is given in Appendix A.

### 4.1. DNA Extraction, PCR Amplification, and DNA Sequencing

For the amplification of the cpDNA 6 non-coding (intron) regions and mtDNA 6 non-coding (intron) regions, genomic DNA was isolated and amplified as described in previous study [38].

The sequences analysed in this paper have been deposited in the NCBI Genbank nucleotide sequence database with the accession numbers MH893827-MH894176 [38] (Appendix A).

### 4.2. SPInDel Analyses

The analysis for cpDNA and mtDNA regions were performed independently. The sequence alignments of each family for the four different cpDNA regions (*atpB-rbcL* intergenic spacer, *trnT*(UGU)-*trnL*5′exon intergenic spacer, *trnL*(UAA) intron intergenic spacer and *trnD*[tRNA-Asp(GUC)]-*trnT*[tRNA-Thr(GGU)]) as well as six different mtDNA regions (*nad1* exon B-*nad1* exon C intron intergenic spacer, *nad4/1-2* intergenic spacer, *nad4L-orf25* intergenic spacer, *rps12-1/nad3-2* intergenic spacer, *rps12-2/nad3-1* intergenic spacer and *rrn5/rrn18-1* intergenic spacer) were submitted to the SPInDel workbench in order to perform diverse calculations [81]. 35 species were selected for the assessment of intra-species diversity (Appendix A). The alignments were analysed in the SPInDel workbench using the same conserved regions defined previously for the family of each species. The SPInDel concept is based on the combination of sequence lengths from different genomic regions. Therefore, in order to perform the diverse statistical analyses available on the SPInDel workbench, the alignments were concatenated of different cpDNA and mtDNA regions. The concatenated alignments were exported to the SPInDel workbench and analysed as previously described using the conserved regions defined for the individual regions. In these analyses, we have excluded the hypervariable regions defined by the peripheral conserved region of adjacent targets since they are not close to each other in the cpDNA and mtDNA. Therefore, the obtained profiles are only composed of the hypervariable regions inside each target region.

## 5. Conclusions

The results obtained in this study clearly indicated disproportions in the available information regarding various non-coding cpDNA and mtDNA regions used in phylogenetic studies, and some of them—due to high variability—can be successfully used in the analyses of closely related species.

The results indicate regions that may be useful molecular markers in studies on closely related species of the genus *Secale*. These include the non-coding regions of chloroplast DNA: *atpB-rbcL* and *trnT* (UGU)-*trnL*(UAA)5′exon and non-coding regions of mitochondrial DNA: *nad1*exonB-*nad1*exonC and *rrn5/rrn18-1*.

In general, our method was able to unambiguously discriminate closely related species in well-supported monophyletic clades.

In summary, the SPInDel approach can be used for the identification of *Secale* species, however the chloroplast genome is more informative. These results suggest that this method can be used for taxonomic classification of *Secale* species as long as appropriate conserved regions are selected.

## Figures and Tables

**Figure 1 ijms-21-09421-f001:**
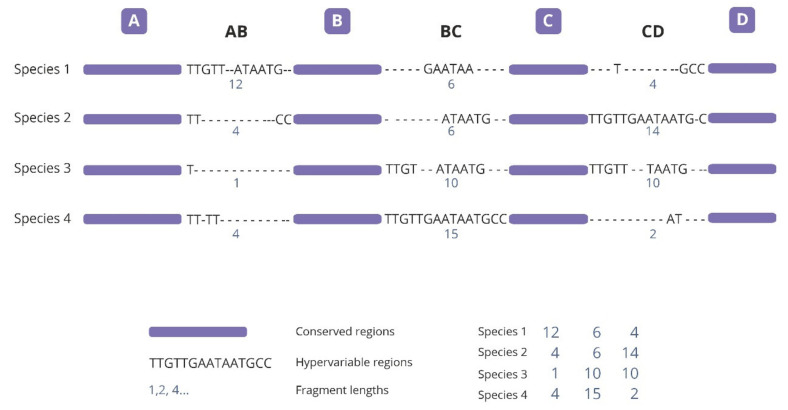
Schematic illustration presenting the strategy used in the species identification by the insertions/deletions method (SPInDel). Illustration shows the sequence alignment for four hypothetical species (1 to 4). Three hypervariable domains (dotted lines) are defined by four conserved regions (blue). The compartent of the alignment is developed to indicate the presence of a large number of gaps in hypervariable regions. Subsequent species are identified using the numerical profile caused by the combination of lengths in the hypervariable regions (blue numeric codes).

**Figure 2 ijms-21-09421-f002:**
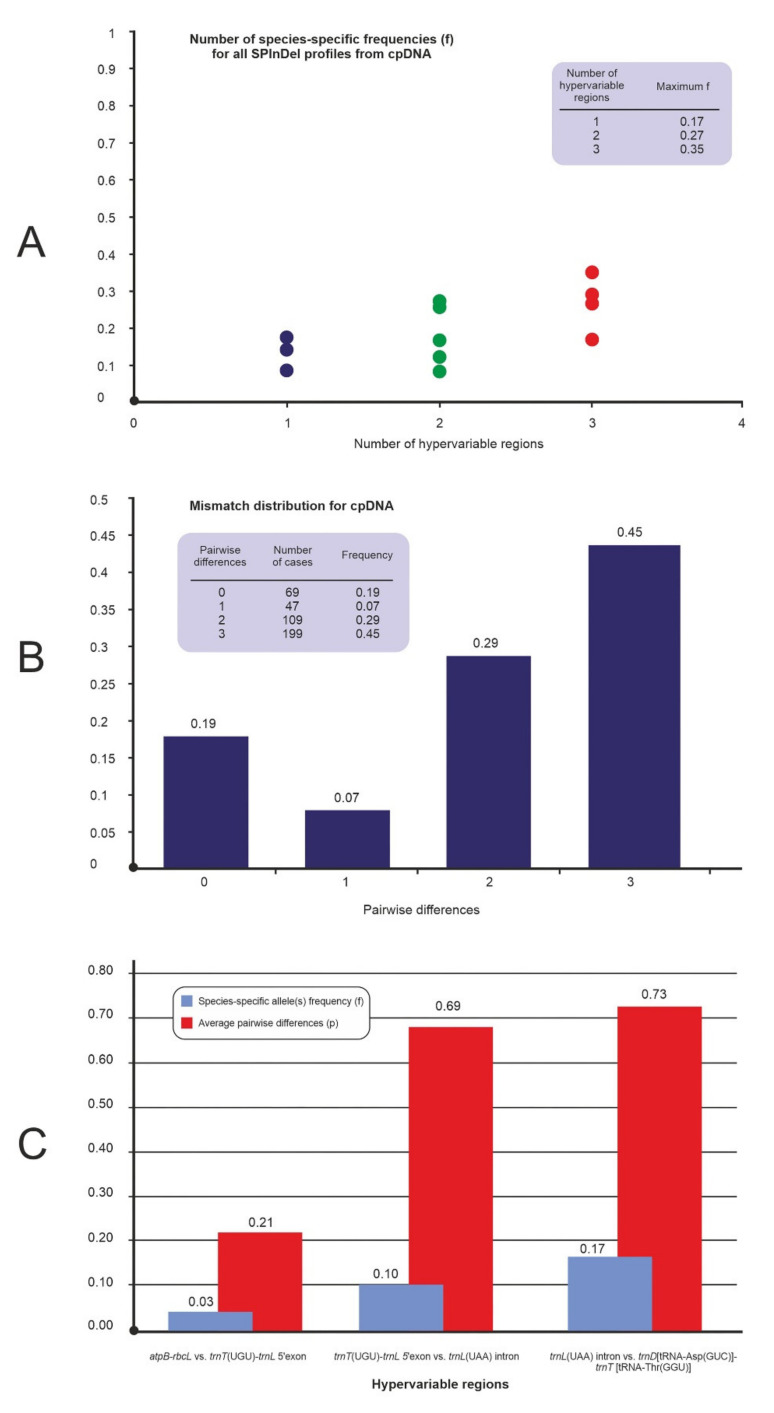
SPInDel analysis of 4 cpDNA *variable-length sequences*; (**A**) The frequency of species-specific profiles in all combinations of hypervariable cpDNA regions; (**B**) Mismatch distribution, i.e., the frequency distribution of the number of SPInDel hypervariable cpDNA regions that differ between all pairs of SPInDel profiles in a taxonomic group; (**C**) The discriminatory potential of each hypervariable cpDNA region individually (region by region analyses).

**Figure 3 ijms-21-09421-f003:**
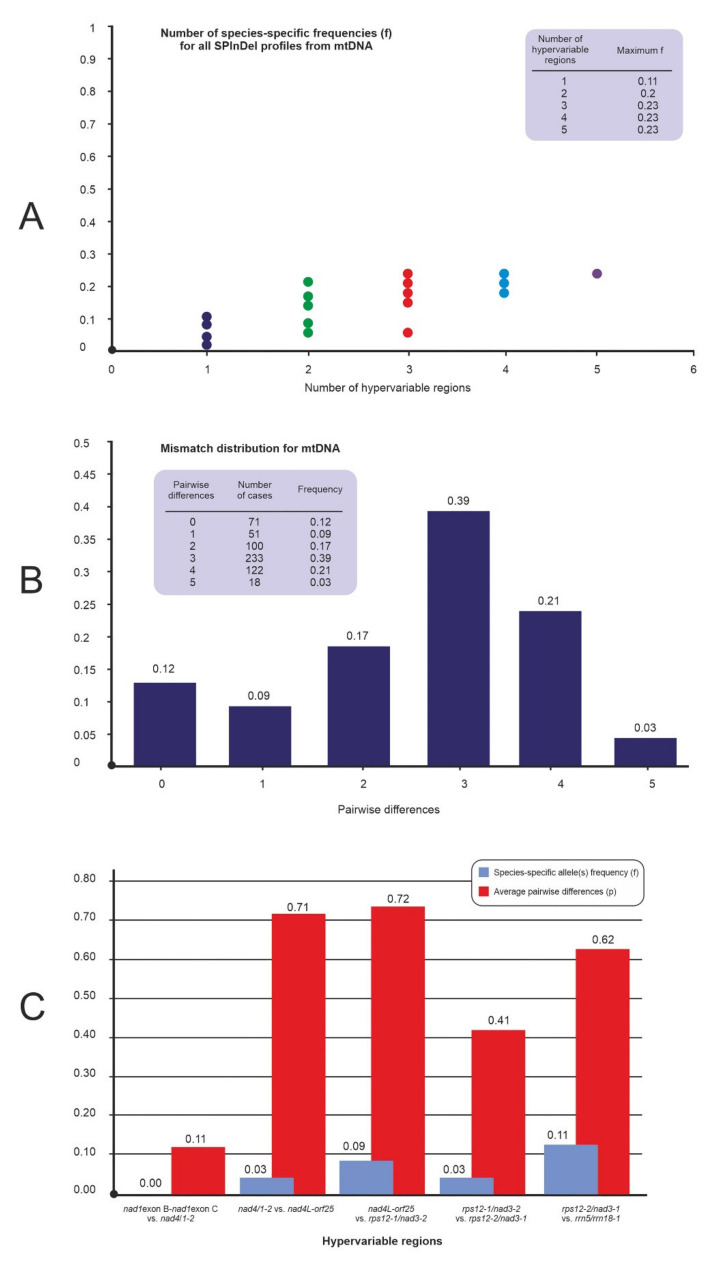
SPInDel analysis of 6 mtDNA noncoding regions; (**A**) The frequency of species-specific profiles in all combinations of hypervariable mtDNA regions; (**B**) Mismatch distribution, i.e., the frequency distribution of the number of SPInDel hypervariable mtDNA regions that differ between all pairs of SPInDel profiles in a taxonomic group; (**C**) The discriminatory potential of each hypervariable mtDNA region individually (region by region analyses).

**Table 1 ijms-21-09421-t001:** Main SPInDel analyses performed for each cpDNA and mtDNA target region for 35 accessions of 13 *Secale* species.

Genomic Region	Number of Conserved Regions	Number of Hypervariable Regions (*n*)	Average Number of Pairwise Differences (p^G^_n_)	Average Number of Pairwise Differences Per Hypervariable Region	Number of Species-Specific Profiles (Nsp)	Frequency of Species-Specific Profiles (f_sp_)	Number of Species-Shared Profiles	Number of Minimum Hypervariable Regions for Discrimination of All Species
cpDNA								
*atpB-rbcL* intergenic spacer	6	2	1.02	0.91	1	0.91	4	4
*trnT*(UGU)-*trnL*(UAA)5′exon intergenic spacer	5	2	0.97	0.62	2	0.83	4	4
*trnL*(UAA) intron intergenic spacer	4	2	0.51	0.41	4	0.87	2	1
*trnD*[tRNA-Asp(GUC)]-*trnT*[tRNA-Thr(GGU)] intergenic spacer	4	1	0.07	0.02	3	0.72	3	2
mtDNA								
*nad1*exon B-*nad1*exon C intron intergenic spacer	4	1	0.06	0.01	2	0.89	2	2
*nad4/1-2* intergenic spacer	4	2	0.27	0.23	1	0.72	2	2
*nad4L-orf25* intergenic spacer	5	2	1.23	0.99	7	0.45	4	3
*rps12-1/nad3-2* intergenic spacer	4	1	0.07	0.05	2	0.67	2	1
*rps12-2/nad3-1* intergenic spacer	3	1	0.01	0.01	1	0.81	1	1
*rrn5/rrn18-1* intergenic spacer	5	2	0.73	0.23	3	0.86	2	2

cpDNA—chloroplast DNA; mtDNA—mitochondrial DNA.

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
