# Peer review of "SPInDel Analysis of the Non-Coding Regions of cpDNA as a More Useful Tool for the Identification of Rye (Poaceae: Secale) Species"

_ijms, 2020, doi:10.3390/ijms21249421_

Round 1

Reviewer 1 Report

The article: „SPInDel analysis of the non-coding regions of cpDNA as a more useful tool for the identification of rye (Poaceae: Secale) species “ is well written and performed study. However I would suggest some minor changes.

In L 60 – explain GWAS

In L 217 and L 222: you describe species – specific profiles as fsp and in the Table 1 as fgn – please be consistent.

L 258- change whitch to which.

Author Response

I would like to read acronyms of the variable regions during the description.

We have added the acronyms in the abstract section.

Line 49: reference 15 is before reference 14 at line 51. Please, check the references order.

Thank you for pointing this out. We have corrected the references order throughout whole article.

Lines 191-194: "However, in a few recent works a high level of species discrimination is attainable in all taxa of life by...".  References to the "recent works" are missing.

Thank you for pointing this out. We have added new reference.

Line 196: please, revise this sentence.

As suggested by the reviewer, the sentence was changed.

Reviewer 2 Report

I have reviewed the manuscript "SPInDel analysis of the non-coding regions of cpDNA as a more useful tool for the identification of rye (Poaceae: Secale) species" that describes the analysis of variable regions in both cpDNA and mtDNA. The manuscript is well written. Introduction is clear and results are well described. The discussion is complete even if some improvement could be done.

I suggest for the publication after minor revisions following the minor comments:

Minor comments:

I would like to read acronyms of the variable regions during the description.

Line 49: reference 15 is before reference 14 at line 51. Please, check the references order.

Lines 191-194: "However, in a few recent works a high level of species discrimination is attainable in all taxa of life by...".  References to the "recent works" are missing.

Line 196: please, revise this sentence.

Author Response

In L 60 – explain GWAS

As suggested by the reviewer, we have added explanation

In L 217 and L 222: you describe species – specific profiles as fsp and in the Table 1 as fgn – please be consistent.

Thank you for pointing this out. We have corrected all abbreviations throughout the article.

L 258- change whitch to which.

Thank you for pointing this out. This sentence refers to our previous article. It has been corrected.